# Jamming and unusual charge density fluctuations of strange metals

Stephen J. Thornton [1], Danilo B. Liarte [1,2,3], Peter Abbamonte [4], James P. Sethna[1] & Debanjan Chowdhury [1]

The strange metallic regime across a number of high-temperature super-conducting materials presents numerous challenges to the classic theory of Fermi liquid metals. Recent measurements of the dynamical charge response of strange metals, including optimally doped cuprates, have revealed a broad, featureless continuum of excitations, extending over much of the Brillouin zone. The collective density oscillations of this strange metal decay into the continuum in a manner that is at odds with the expectations of Fermi liquid theory. Inspired by these observations, we investigate the phenomenology of bosonic collective modes and the particle-hole excitations in a class of strange metals by making an analogy to the phonons of classical lattices falling apart across an unconventional jamming-like transition associated with the onset of rigidity. By making comparisons to the experimentally measured dynamical response functions, we reproduce many of the qualitative features using the above framework. We conjecture that the dynamics of electronic charge density over an intermediate range of energy scales in a class of strongly correlated metals can be at the brink of a jamming-like transition.

A hallmark of numerous interacting phases of quantum matter are their long-lived collective excitations (such as phonons, magnons, and skyrmions). Microscopically, these collective modes require a coherent motion of the constituent particles in the system. While such modes often have a long lifetime at low energies, they are prone to decay once they encounter the multi-particle continuum at high energies. Even in weakly interacting metals, there are two kinds of long-lived excitations–the plasmon, which represents a collective (longitudinal) density fluctuation, and single-electron like quasiparticle excitations near the Fermi surface. The plasmon eventually decays at large enough momentum and frequency (i.e., for $\omega > \omega_\star(q)$) into the multi-particle continuum due to purely kinematic reasons. Within Landau's original formulation of Fermi liquid (FL) theory for electrically neutral fermions (e.g., as in liquid Helium-3)[1], the zero-sound mode is associated with a collective oscillation of the entire Fermi surface and has properties that are qualitatively similar to a longitudinal acoustic phonon. The sound mode gets renormalized into the plasmon mode in the presence of Coulomb interactions. It is natural to consider the fate of collective modes and their possibly unconventional decay into multi-particle continua in the regime of strong interactions.

Recent advances in the experimental technique of momentum-resolved electron energy-loss spectroscopy (M-EELS)[2] have made it possible to measure the dynamical charge response of numerous strongly correlated materials over a broad range of frequencies and momenta[3–5]. Focusing specifically on the strange metal regime of a cuprate material (BSCCO), these experiments report evidence of a featureless particle-hole continuum extending over most of the Brillouin zone (BZ), while being independent of temperature and doping. Remarkably, the unconventional continuum persists up to the highest measurable energies and accounts for more than 99% of the total spectral weight in the $f$-sum rule[3,4]. Perhaps the most noteworthy observation is the absence of a sharply dispersing plasmon in the BZ (except for a narrow range of momenta, $q \lesssim 0.05$ r.l.u., near the

[1]Department of Physics, Cornell University, Ithaca, NY 14853, USA. [2]ICTP South American Institute for Fundamental Research, São Paulo, SP, Brazil. [3]Institute of Theoretical Physics, São Paulo State University, São Paulo, SP, Brazil. [4]Department of Physics, University of Illinois at Urbana-Champaign, Urbana, IL 61801, USA. ✉e-mail: debanjanchowdhury@cornell.edu

Γ – point[6,7]), as it decays into the featureless continuum. Evidence for such a continuum has been reported in earlier Raman studies[8,9] and recent M-EELS measurements in other strongly interacting metals (e.g., $Sr_2RuO_4$[5]). The microscopic origin for the decay of the plasmon into such continua remains unclear. Recent theoretical works have utilized solvable lattice electronic models[10] to analyze the unconventional particle-hole continuum[11] and the anomalous decay of plasmons[12] in the strongly correlated regime of certain non-Fermi liquid metals; see ref. 13 for a complementary holographic computation of plasmon decay.

In addition to the anomalous dynamics of the charge-density fluctuations, the normal metallic state across a number of strange metals exhibits universal scattering lifetimes[14-17] and violates the Mott-Ioffe-Regel limit with increasing temperature[18,19], suggesting an absence of electronic quasiparticles with a long mean-free path and lifetime. A satisfactory theoretical explanation for the complex and universal aspects of this phenomenology does not presently exist starting from microscopic models.

These results point to the intriguing possibility of the strongly interacting electron fluid forming a collective and self-organized, nearly jammed state. At intermediate energy scales, it is conceivable that certain aspects of the dynamical response associated with the collective modes can be understood by drawing analogies to a strongly correlated classical liquid. By analyzing the universal behavior of such a liquid near the onset of rigidity (Fig. 1a) and a detailed comparison to recent M-EELS experiments in cuprates, we conjecture that the intermediate-scale charge-dynamics in strange metals belongs in the family of a class of theories with critical rigidity correlations. This brings together a new class of problems under the umbrella of jamming, which includes rigidity transitions observed in colloids and granular materials[20,21], living tissues[22,23], elastic networks[24-27], dislocation systems[28,29], deep learning[30], and analogs of metal-insulator transitions for interacting quantum bosons[31]. Quenched randomness in geometrically frustrated magnets has also been shown to produce a jammed spin liquid[32], which is known to display unconventional spin-dynamics[33].

In this manuscript we address the question of what phenomenon might give rise to a largely momentum-independent continuum such as that observed in M-EELS experiments. We conjecture that these observations might be connected to phenomena characteristic of the rigidity transition in granular media[20]. Near such a transition, the vibrational density of states develops an anomalous, nearly frequency-independent plateau[34,35]. This paper will be concerned with addressing the similarities between the experimentally measured density correlations of strange metals and the calculated density correlations near the onset of rigidity, based on our recent analysis of the density response near a rigidity transition[36].

## Results

The onset of rigidity in classical liquids (but without any long-range crystalline order) has a complex dynamical signature. The transition is associated with a singular rearrangement of the low-energy vibrational spectrum of the nearly rigid solid[20]; see Fig. 1b. These low-energy excitations will become the analog of the unconventional particle-hole continuum in the strange metal that we described above. Moreover, the longitudinal phonons in these viscoelastic systems can decay into

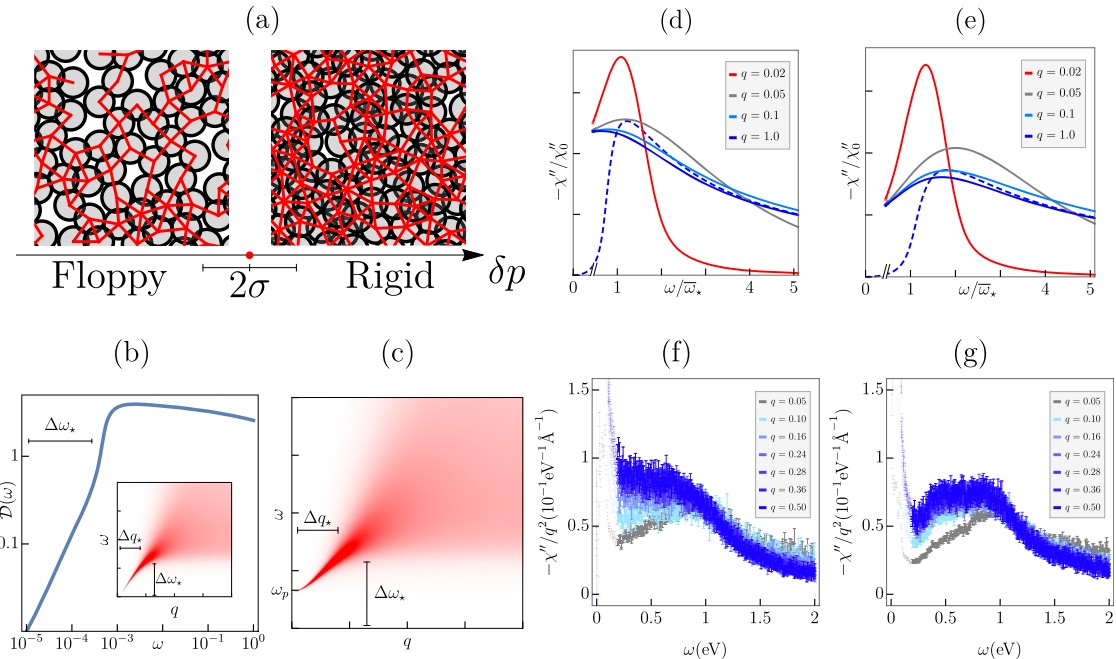

**Fig. 1 | Charge-density response near onset of rigidity and in strange metals. a** A random network of bonds (red) displayed in a rigid vs. floppy system, on either side of a continuous rigidity percolation (RP) transition; the critical properties near RP are distinct from a jamming transition associated with random packings of hard spheres[20]. We hypothesize that the two-particle density response over a broad range of intermediate energies near the hole-doping induced transition associated with the electrons near optimal doping in cuprates[38] can be described as a rigidity-type transition. **b** The vibrational density of states, $\mathcal{D}(\omega)$, as a function of frequency ($\omega$) at a fixed distance from the RP critical point $\delta p = 10^{-3}$. The plateau in $\mathcal{D}(\omega)$ onsets for $\omega > \Delta\omega_\star \sim |\delta p|$. Inset: The polarization function, $\Pi''(q, \omega)$, in the absence of Coulomb interactions, as a function of $q$ and $\omega$, revealing the acoustic collective mode and its damping inside the continuum. **c** The density response function, $\chi''(q, \omega)$, (including the Coulomb interaction) with $\omega_p = 0.66|\delta p|$. The response functions, $\chi''(q, \omega)$, averaged over a range of $|\delta p| \leq \sigma$ (see **a**) as a function of $\omega$ for different $q$ at a fixed distance ($\delta p$) from RP for **d** $|\delta p| = 1.1 \times 10^{-3}$, $\sigma = 0.9 \times 10^{-3}$, and **e** $|\delta p| = 1.6 \times 10^{-3}$, $\sigma = 0.9 \times 10^{-3}$. The plasma frequency is chosen to be at $\omega_p = 0.5\overline{\omega}_\star$. The dashed line represents the $q$-independent shape of the imaginary part of the susceptibility in the absence of disorder. Frequencies in **d** and **e** are rescaled by the same scaling frequency $\overline{\omega}_\star$ associated with the average distance to the transition $\delta p$; see the Methods section for details. Experimental data from M-EELS[3] demonstrating the overall $q$-independent shape of the continuum for **f** optimally doped BSCCO and **g** overdoped BSCCO. Error bars represent statistical (Poisson) error. The lowest frequencies show the lattice phonon, which we do not describe in our framework.

this continuum of low-energy vibrational excitations, much like the plasmons do in the cuprate strange metal.

Starting with scaling forms for the longitudinal susceptibility that were derived recently by some of us[36,37], we will write down a coarse-grained effective description for the long-wavelength and low-frequency bosonic excitations in a liquid at the brink of rigidity percolation (RP); see Fig. 1a. Percolation is a transition in connectivity; rigidity percolation is a transition from elastic to floppy, with a dramatic peak in low-energy excitations we believe common to strange metals. We will extend our formalism to analyze the inelastic density-density response using the predictions of our scaling theory and make direct comparisons with the M-EELS results, highlighting the similarities between the mechanism for anomalous decay of the plasmon into the continuum at momenta away from the $\Gamma-$ point. Given the relatively large energy-scales over which the charge response has been probed, it is likely that the quantum critical collective modes associated with various forms of broken symmetries that emerge at low-energies[38] do not play a fundamental role in the interpretation of the M-EELS experiments.

Our starting point is based on a recently proposed scaling ansatz for the dynamical susceptibilities near classical jamming and RP[36,37]. There has been a dearth of solvable models in finite dimensions where universal features of the dynamical susceptibilities can be analyzed in a reliable fashion; we utilized the tractable effective medium theory[26] to compute these in refs. 36, 37 and obtained their explicit analytical forms. Given that the strange metals where the anomalous density fluctuations have been observed are quasi two-dimensional, we will model our system as a stack of weakly coupled two-dimensional layers. The individual layers are described in terms of a randomly percolated lattice of harmonic springs (Fig. 1a); the connection to the density fluctuations of an underlying electronic fluid will be made explicit later. For our present discussion, we will start specifically with the longitudinal part of the displacement response, $\Xi_L$, near RP,

$$\Xi_L(q,\omega) \approx -|\delta p|^{-\gamma} \mathcal{L}(\widetilde{q}, \widetilde{\omega}), \tag{1a}$$

$$\widetilde{q} \equiv \frac{q}{|\delta p|^{\nu}}, \quad \widetilde{\omega} \equiv \frac{\omega}{|\delta p|^{z\nu}}, \tag{1b}$$

where $q$ and $\omega$ represent the wavevector and frequency, respectively, and $|\delta p|$ represents the deviation away from the critical point. The critical exponents for susceptibility, correlation length, and correlation time are denoted $\gamma$, $\nu$, and $z$, respectively. For RP, our calculation leads to $\gamma = 2, z = 2$ and $\nu = 1/2$. In two-dimensions, the above scaling form has additional dependence on the logarithms of the scaling variables which do not qualitatively affect any of our results; a detailed discussion of the origin of these additional logarithms will be discussed elsewhere (see Sec. I of the Supplementary Information[39] for more details). $\mathcal{L}(\widetilde{q}, \widetilde{\omega})$ is a universal scaling function whose explicit form appears in the Methods section. In all of our subsequent analysis and in our comparison with the experimental results, $\Xi_L(q,\omega)$ will play a central role. Near RP, the transverse response, $\Xi_T(q,\omega)$, has the same universal scaling form as $\Xi_L(q,\omega)$ but with different non-universal constants.

The onset of rigidity is tied to a significant rearrangement of the vibrational density of states, $\mathcal{D}(\omega)$; see Fig. 1b and Methods for a definition. Near RP, $\mathcal{D}(\omega) \sim \omega$ for $\omega \lesssim \Delta\omega_\star \cdot |\delta p|$ (up to additional logarithms). For $\omega \gtrsim \Delta\omega_\star, \mathcal{D}(\omega)$ has a remarkably flat continuum as a function of $\omega$ over several orders of magnitude of frequencies; see Fig. 1b. The physical origin of this low-energy continuum is related to the boson peak that demarcates a crossover from Debye to isostatic behavior, and is a recurring feature in the physics of glassy systems[35,40,41]. From the point of view of our analogy to the excitations in the strange metal, these modes are naturally interpreted as the particle-hole continuum. This analogy will become more direct when we analyze the nature of the collective excitations—these are the phonons of the solid becoming floppy, which turn into the plasmon in the strange metal with the inclusion of Coulomb interactions—and their decay into the flat $\mathcal{D}(\omega)$ near RP.

In order to make the analogy between classical liquids and their vibrational excitations to the collective modes in strange metals, we need a precise relationship between the longitudinal susceptibility ($\Xi_L$) and the electron density correlation functions. As in the jellium model, we assume the negatively charged electronic liquid co-exists with a uniform oppositely charged (static) background to maintain electrical neutrality; we are only interested in the dynamics of the former. In the proposed model, the changes in the local displacement, $\mathcal{U}$, are tied to a local fluctuation of the electronic number density. More precisely,

$$n(\boldsymbol{x}) = n_0(1 - \nabla \cdot \mathcal{U}), \tag{2}$$

where $n_0 = \rho/m$ is the average background density. One of the central quantities of interest is the polarization function, $\Pi(q,\omega) = n_0^2 q^2 \, \Xi_L(q,\omega)$, which is related to the longitudinal susceptibility introduced earlier. This is the density-density response of the neutral system near the transition. Since the electronic liquid is charged and interacts via repulsive Coulomb interactions, $V(|\boldsymbol{x} - \boldsymbol{x}'|)$, we include it explicitly as

$$\Delta U = \frac{1}{2} \int_{\boldsymbol{x}} \int_{\boldsymbol{x}'} \delta n(x) V(|\boldsymbol{x} - \boldsymbol{x}'|) \delta n(x'), \tag{3}$$

where $\delta n(\boldsymbol{x}) = n(\boldsymbol{x}) - n_0 = -n_0 \nabla \cdot \mathcal{U}$. The experimentally measured density-density response, $\chi(q,\omega)$, can be obtained from the polarizability after including the effects of Coulomb interactions,

$$\chi(q,\omega) = \frac{\Pi(q,\omega)}{1 - V(q)\Pi(q,\omega)}. \tag{4}$$

In the remainder of this study, we will calculate $\chi(q,\omega)$ near RP using the universal form of $\Xi_L(q,\omega)$, and highlighting both its similarities and differences when compared against the experimentally measured density response function in the cuprate strange metal. See Sec. II of the Supplementary Information[39] for more details.

To analyze the effect of the plasmon decay into the continuum, it is conceptually simpler to approach the transition from the rigid side. The imaginary part of the susceptibility, $\chi''(q,\omega)$, reveals a sharply dispersing plasmon for $\Delta q_\star \sim |\delta p|^{1/2}$ (up to logarithms), controlled by the distance to RP ($\delta p$), that broadens significantly as a result of decay into the low-energy vibrational states over a broad range of wavevectors and frequencies; see inset of Fig. 1b. The effect of $V(q)$ on $\chi(q,\omega)$ is to renormalize the acoustic mode to the plasma frequency, $\omega_p = \sqrt{4\pi e^2 n_0/m}$, where we have assumed the three-dimensional form, $V(q) = 4\pi e^2/q^2$; see Fig. 1c. The broadening of the plasmon due to decay into the unconventional continuum remains identical. The phenomenology described here is exactly what we set out to achieve theoretically inspired by the M-EELS experiments in strange metals—a plasmon that is damped beyond small momenta $q \gtrsim \Delta q_\star$ into a featureless, low-energy continuum. The close similarity that we demonstrate between the unconventional decay of the phonon into the vibrational continuum near RP and of the plasmon into the measured particle-hole continuum in strange metals is one of the central results of this paper.

Let us next turn to studying the detailed $q, \omega-$ dependence of $\chi(q,\omega)$ near RP in order to make further comparisons with the measured charge response functions. For the smallest values of $q$, there is a sharp plasmon that appears at the plasma frequency, $\omega_p$. For $q \gtrsim \Delta q_\star \sim |\delta p|^{1/2}$, the plasmon broadens rapidly, and $\chi(q,\omega)$ becomes nearly $q$-independent with a broad feature centered near $\Delta\omega_\star$. Increasing $q$ further serves only to adjust the crossover frequency

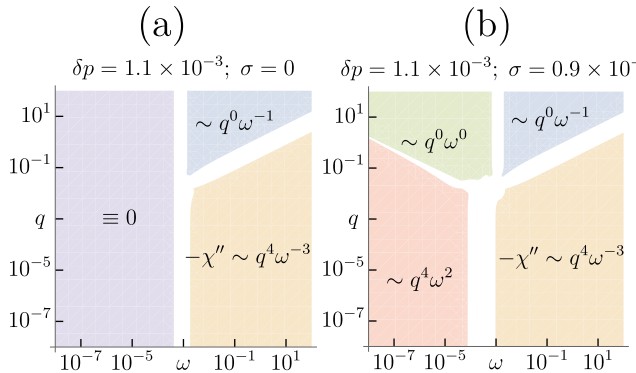

**Fig. 2 | Distinct spectroscopic regimes of the charge-density response.** Frequency and momentum-dependence of $\chi''$, **a** without ($\sigma = 0$) and **b** with ($\sigma = 0.9 \times 10^{-3}$) disorder averaging. Here $|\delta p| = 1.1 \times 10^{-3}, e = 10^{-4}$. In **a**, the largest values of $q$ lead to a bump in the susceptibility at a $q$-independent frequency $\Delta\omega_\star$, followed by a decay $\sim\omega^{-1}$ eventually crossing over into $\sim\omega^{-3}$ Drude-like behavior. In **b**, a plateau in the response emerges at the lowest frequencies whose width is set by $\sigma$. The qualitative behavior is retained even after including corrections to the response that fix the scaling in the lowest frequency regime (see Fig. 1d, e).

beyond which there is a crossover to a $1/\omega^3$ falloff, in accordance with the $f$-sum rule (see Fig. 2a). The $q$-independent shape of $\chi''(q, \omega)$ is also shown as the dashed blue curve in Fig. 1d, e. This broad feature is tied to the same boson peak that was discussed above in the context of the onset of the enhancement of the low-energy modes in $\mathcal{D}(\omega)$.

Although our form of $\chi''(q,\omega)$ near the transition reproduces the strongly overdamped plasmon and the $q$-independent shape of the response over the measured frequency range, the response at the lowest frequencies does not have the characteristic plateau of the experiment. To address the possible origin of this feature, we can appeal to the inherent inhomogeneity that is present in these materials. There is experimental evidence for nanoscale electronic inhomogeneity across multiple families of cuprate single crystals (including, e.g., BSCCO)[42,43]. For a given sample at a fixed nominal doping, the experiments probe the density response averaged over all of the inhomogeneous regions of the sample. To replicate this feature in our theoretical analysis, we sample and smear our results for $\chi(q, \omega)$ over a distribution of $\delta p$. We thus assume that the variations in doping level change the distance to the onset of rigidity. Our averaging presumes the disorder does not couple to the translational Goldstone mode of the transition (by solely changing the density of bonds). The doping, however, breaks translational symmetry, and pinning on defects is also known to lower the threshold of rigidity[44–47]. Adding the effects of pinning to our analysis could be fruitful in future work. The qualitative effects of the above averaging procedure are similar for any smooth, symmetric distribution.

Near the boson peak, the disorder-averaged susceptibility is most drastically altered. When the mean deviation from criticality is comparable to the width of the disorder distribution, $|\delta p| \leq \sigma$, the spectrum becomes dispersionless as a function of $\omega$ for large $q > \Delta q_\star$; see Fig. 1d. Within our framework, the frequency-independent plateau observed near optimal doping can be interpreted as the disorder-induced smearing of the boson peak near RP. Beyond this featureless region, there is a crossover into an anomalous power-law regime, $\chi''(q, \omega) \sim 1/\omega^\alpha$ with $\alpha < 3$. Both of these features are similar to the experiments[3]; see Fig. 1d–g for a comparison. At the largest frequencies, the asymptotic forms of the polarization with and without disorder-averaging are identical with $\alpha = 3$. The shape of the susceptibility is largely independent of $q$ over a wide range of $\omega$. This leads to a $q$-independent crossover frequency from the plateau to a power-law falloff at large $\omega$. As we move away from the transition fixing the magnitude of the disorder $\sigma$, the plateau at low frequencies evolves into a bump; see

Fig. 1e. This bump can be interpreted as a severely overdamped plasmon, whose location becomes nearly $q$-independent at large $q$. The $q$-independence is tied to the decay into the particle-hole continuum, whose onset is at a fixed $\Delta\omega_\star$.

The power-law scaling behavior of the singular part of the susceptibility $\chi''$ before and after the inclusion of disorder averaging is illustrated in Fig. 2. At the highest frequencies, the power-law scaling is unaffected by the specific type of disorder considered here. At low frequencies, we see the emergence of a plateau region whose width is $q$-independent (and set by the amount of disorder $\sigma$) for $q \gtrsim \Delta q_\star$. For experimental measurements close to this critical point, all wavevectors except for those closest to the center of the BZ will probe the incoherent plateau rather than the collective mode. The most notable difference between this framework and the one observed in the experiments is in the wavevector dependence of the magnitude of the response. If the response has a $q$-independent shape at all frequencies, then one infers that it must scale as $\sim q^2$ to satisfy the $f$-sum rule. The singular responses computed in this paper also satisfy the appropriate sum rules, since $q$ sets the frequency at which we crossover into the Drude-type scaling $\sim\omega^{-3}$. See Sec. III of the Supplementary Information[39] for more details. A recent complimentary theoretical work[12] finds a distinct high-frequency scaling $\sim 1/\omega^2\log^2(\omega)$, which is also consistent with the $f$-sum rule and is in better qualitative agreement with the experiments.

Within the framework of rigidity percolation, we have pointed out an intriguing analogy between the large collection of low-energy vibrational modes and the particle-hole continuum of strange metals, into which collective modes can rapidly decay. The onset energy of this decay is set by the distance to the critical point. Although the details of the specific momentum-dependence for the polarizability are not in perfect agreement with the MEELS experiment, we can reproduce a $q$-independent shape for $q > \Delta q_\star$ that is set by the distance to the critical point. It is possible that a different, and yet to be understood, universality class of rigidity transition displays a power-law density response that agrees better with the experiments. A broad implication of our hypothesis is that over a range of intermediate energy scales over which the density correlations in strange metals appear to display features like jamming, the electronic fluid might also display interesting memory effects known to arise in glassy systems and near rigidity transitions. Finding new experimental ways to probe this physics remains an interesting future direction. Developing a microscopic quantum theory of interacting electrons whose effective theory reduces to an analogous rigidity transition is a challenging open problem[31,48]. In this regard, exploring possible connections between the low-energy vibrational excitations near jamming and the low-energy non-quasiparticle-like excitations in the solvable quantum Sachdev-Ye-Kitaev models[10,49] will be an interesting theoretical exercise.

## Methods
### Universal scaling function
In three spatial dimensions and higher, the scaling function $\mathcal{L}(\widetilde{q}, \widetilde{\omega})$ has the explicit form[36,37]

$$\mathcal{L}(\widetilde{q}, \widetilde{\omega}) = \left[ a\widetilde{q}^2 \mathcal{M}_\pm(\widetilde{\omega}) - \widetilde{v}(\widetilde{\omega}) \right]^{-1}, \tag{5a}$$

$$\mathcal{M}_\pm(\widetilde{\omega}) = b \left[ \sqrt{1 - c\,\widetilde{v}(\widetilde{\omega})} \pm 1 \right], \tag{5b}$$

where $a$, $b$ and $c$ are constants, with $\widetilde{v}(\widetilde{\omega}) = \rho\,\widetilde{\omega}^2$ and $i\gamma\widetilde{\omega}$ for undamped and overdamped dynamics, respectively. Here, $\rho$ is a mass density and $\gamma$ is a viscous drag coefficient. The plus and minus signs in $\mathcal{M}_\pm(\ldots)$ correspond to the rigid and floppy states, respectively. We use the undamped form of the response exclusively.

## Vibrational density of states

The vibrational density of states can be computed from

$$\mathcal{D}(\omega) = -\frac{\omega}{\pi}\int_{BZ} d^2q\, \text{Tr}\left(\text{Im}\left[\mathcal{G}_{ij}(q,\omega)\right]\right), \tag{6a}$$

$$\mathcal{G}_{ij}(q,\omega) = \Xi_L(q,\omega)\hat{q}_i\hat{q}_j + \Xi_T(q,\omega)\left(\delta_{ij} - \hat{q}_i\hat{q}_j\right) \tag{6b}$$

The density of states has an additional $\omega$ prefactor as we are considering excitations in a classical disordered system.

## Disorder averaging of charge-density response

We convolve our universal scaling function with a specific disorder distribution, such that the effective disorder-averaged polarization function takes the form (denoted '·')

$$\overline{\chi_{\delta p}(q,\omega)} = \int_{-\infty}^{\infty} d(\Delta p')\mathcal{P}_\sigma(\Delta p')\chi_{\delta p'}(q,\omega) \tag{7a}$$

$$\mathcal{P}_\sigma[\Delta p'] = \frac{1}{\sqrt{2\pi\sigma^2}}e^{-(\Delta p')^2/2\sigma^2}, \tag{7b}$$

where $\chi_{\delta p}(q,\omega)$ is the response at a fixed distance ($\delta p$) from RP, and we choose a Gaussian distribution, $\mathcal{P}_\sigma[\Delta p]$, with width $\sigma$ and $\Delta p' \equiv \delta p' - \delta p$. For the forms of the response in Fig. 1d, e, the frequency is rescaled by $\overline{\omega}_\star = 10^{-3}$ in both figures, instead of the distinct $\delta p$. This is to make comparison to the experiment, where the dopant concentration is changed (moving the system further from a critical point in our framework), but not the frequency scale.

## Data availability

The experimental MEELS data for BSCCO used to generate Fig. 1f–g have been publicly available on *Zenodo* since the publication of ref. 3 (at ref. 50). These figures are a reproduction of Figs. 2a and 4a in ref. 3 without the offset in $q$.

## Code availability

All theoretical plots were made in Mathematica, and the associated notebooks are available on *Zenodo* (at ref. 51).

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

## Acknowledgements

P.A. thanks Vincenzo Vitelli for suggesting a possible connection between jamming and the measured density response in Fig. 1. This work was supported in part by NSF DMR-1719490 (S.J.T. and J.P.S.). D.C. is supported by faculty startup funds at Cornell University. P.A. acknowledges support from the EPiQS program of the Gordon and Betty Moore Foundation, grant no. GBMF9452. DBL thanks financial support through FAPESP grants # 2016/01343-7 and # 2021/14285-3.

## Author contributions

P.A. provided the initial idea for the problem, contributed experimental insight, and provided the experimental data. J.P.S. and D.C. designed and supervised the research. S.J.T. and D.B.L. carried out the research. S.J.T. and D.C. wrote the manuscript, with input from all authors.

## Competing interests

The authors declare no competing interests.
