## [Peer Review File · Nature Communications]

REVIEWER COMMENTS

Reviewer #1 (Remarks to the Author):

Review for

Jamming and Unusual Charge Density Fluctuations of Strange Metals

S. Thornton et al

This paper addresses the behavior of certain metallic systems that exhibit significantly different properties compared to the Fermi Liquids. There are a large number of these systems that exhibit such behaviors and can provide representations of various ideas from quantum critical points, correlation effects, Bosons that are not in condensates and so forth. The authors explore the use a framework from jammed systems or systems close to jamming. Jamming as originally proposed occurs for systems of grains, or particles that are fluidlike with weak correlations below a certain density but become solidlike above some density and develop longer range correlations. The jamming transition itself also has interesting features. The work focuses on lifetimes of excitations which are long lived in pure phases but decay rapidly near a jamming transition. The framework also matches with several observations. There have been a few theory works in which there are discussions about possible jamming-like criticality in electronic systems, but this paper goes much beyond that in that the authors provide a much clearer picture of how the jamming would work and how it matches with some observations.

In principle this sounds very interesting and is of broad interest, but as was the case for some of the previous work on ideas of jamming in metals or charge ordering states, I have some conceptual issues that I would like the authors to clarify. I also think such clarification will make this work more appealing to both the jamming and charge ordering communities.

(1) I am bit confused about how is this different from 2D metal insulator transitions, which also appear to show some kind of transition that is perhaps of the directed percolation type. Some ideas are that the metal-insulator transitions involve localized charged fluid regions that percolate above some density. The jamming transition might also be thought of as a percolation or directed percolation transition, since below the threshold there is a short range decay, well above the threshold there is a long range decay, and at the transition there is some intermediate decay in the correlations. The authors should clarify this point.

(2) Related to point (1), in systems with jamming one can typically tune the transition by varying the particle density. What is the equivalent to a tunable parameter in the systems considered in the present work? If such tunability were achievable experimentally, I would expect that below the transition the system would show properties of an insulator or some other type of metal. Is there a reason that only a certain class of systems seems to fall just at the jamming transition, and can these systems be tuned?

(3) Jamming and the glass transition actually occur at very different temperatures and densities in particle like systems. Is this also the case in this system?

(4) One thing that I think could make this picture a bit different is that charge states are also coupled to a substrate that can provide quenched disorder. It is known that quenched disorder or pinning can change the jamming transitions by adding a characteristic cutoff length or changing the location of the transition points. It might be possible that if there is additional quenched disorder, some parts of the system will naturally be near the jamming transition, or there could be an extension of the regime where criticality associated with jamming can occur. The authors should add some comments on this and the possible role of disorder or heterogeneity in these samples and how jamming can still work. Additionally,

since such disorder combined with jamming or clogging can produce strong memory effects, do these strange metals also have memory effects or fragility? This might be a good prediction as well. Some related papers on the particle systems are:

Jamming in systems with quenched disorder

C. J. Olson Reichhardt, E. Groopman, Z. Nussinov, and C. Reichhardt

Phys. Rev. E 86, 061301 (2012)

Pinning Susceptibility: The Effect of Dilute, Quenched Disorder on Jamming

Amy L. Graves, Samer Nashed, Elliot Padgett, Carl P. Goodrich, Andrea J. Liu, and James P. Sethna

Phys. Rev. Lett. 116 235501 (2016)

Crossover from Jamming to Clogging Behaviours in Heterogeneous Environments

H. Péter, A. Libál, C. Reichhardt, and C. J. O. Reichhardt

Sci Rep 8 10252 (2018)

Jammed solids with pins: Thresholds, force networks, and elasticity

Andy L. Zhang, Sean A. Ridout, Celia Parts, Aarushi Sachdeva, Cacey S. Bester, Katharina Vollmayr-Lee, Brian C. Utter, Ted Brzinski, and Amy L. Graves

Phys. Rev. E 106 034902 (2022)

(5) Fig. 2 is a nice diagram explaining the different decay exponents, but I think another nice diagram to include would be something like a jamming phase diagram but for the metal case. In that case, what is being tuned, the doping or charge density, the temperature, or other quantities? This would probably get some more interest in the paper from the soft matter community. Overall the connection to soft matter is nice but I would like to know why these materials seem to go to the critical point. The authors could add some comments on this point. For example, it could be due to longer range correlations of the charge that allow the system to be close to some kind of jamming, as found for dislocations:

Dislocations Jam at Any Density

Georgios Tsekenis, Nigel Goldenfeld, and Karin A. Dahmen

Phys. Rev. Lett. 106, 105501 (2011)

(6) Since the authors use arguments from jamming, could this work imply the presence of some other phenomena that is found in jamming, such as the situation where an already jammed system could unjam under a drive, undergo thermal unjamming, or so forth? As I mentioned already, jammed systems can in many cases have memory or fragility effects, particularly near the threshold.

Reviewer #2 (Remarks to the Author):

I very strongly recommend the publication of the manuscript "Jamming and Unusual Charge Density Fluctuations of Strange Metals" by Thornton, Liarte, Abbamonte, Sethna, and Chowdhury. This article

is very well written and motivated. The presented analysis is extremely straightforward and lucid. There are no real required changes at this end prior to publication.

As an extremely minor point, that should not be misconstrued as a condition for acceptance, it might be remarked that under the noted "umbrella of jamming", Ref. [44] already earlier suggested, from a rather different vantage point, the prospect of jamming in interacting electronic systems. Indeed, similarly inspired by the phenomenology of the high temperature superconductors, Ref. [44] illustrated how "self-organized" jamming may be triggered by particle interactions.

The manuscript by Thonton, Liarte, Abbamonte, Sethna, and Chowdhury advances a conceptually neat idea and simple framework that may rationalize the phenomenology of bad metals. This is important and may be appreciated by numerous readers.

Dear Editors,

We thank the reviewers for their detailed comments. We are glad to see that both reviewers find our work to be interesting. While reviewer #2 has already recommended publication of our manuscript as it is, reviewer #1 has asked a few questions, which we have answered below. In addition, we have also included clarifying remarks in the manuscript (wherever necessary) in light of some of the comments by reviewer #1.

We hope that the revised manuscript will now be accepted for publication in Nature Communications.

Sincerely,

Authors

REVIEWER COMMENTS

Reviewer #1 (Remarks to the Author):

This paper addresses the behavior of certain metallic systems that exhibit significantly different properties compared to the Fermi Liquids. There are a large number of these systems that exhibit such behaviors and can provide representations of various ideas from quantum critical points, correlation effects, Bosons that are not in condensates and so forth. The authors explore the use a framework from jammed systems or systems close to jamming. Jamming as originally proposed occurs for systems of grains, or particles that are fluidlike with weak correlations below a certain density but become solid like above some density and develop longer range correlations. The jamming transition itself also has interesting features. The work focuses on lifetimes of excitations which are long lived in pure phases but decay rapidly near a jamming transition. The framework also matches with several observations. There have been a few theory works in which there are discussions about possible jamming-like criticality in electronic systems, but this paper goes much beyond that in that the authors provide a much clearer picture of how the jamming would work and how it matches with some observations.

In principle this sounds very interesting and is of broad interest, but as was the case for some of the previous work on ideas of jamming in metals or charge ordering states, I have some conceptual issues that I would like the authors to clarify. I also think such

clarification will make this work more appealing to both the jamming and charge ordering communities.

We thank the reviewer for the positive remarks regarding our manuscript, and for finding the work to be interesting and of broad interest. We have answered the comments by the referee below, and included additional clarifying remarks in the manuscript (wherever necessary).

(1) I am bit confused about how is this different from 2D metal insulator transitions, which also appear to show some kind of transition that is perhaps of the directed percolation type. Some ideas are that the metal-insulator transitions involve localized charged fluid regions that percolate above some density. The jamming transition might also be thought of as a percolation or directed percolation transition, since below the threshold there is a short range decay, well above the threshold there is a long range decay, and at the transition there is some intermediate decay in the correlations. The authors should clarify this point.

We thank the reviewer for raising this question. Rather than discussing aspects of the metal-insulator transition in the text (where there is indeed a lot of overlapping discussion with bond-percolation), we have clarified in the updated manuscript how rigidity percolation (RP) differs from regular bond percolation. The second paragraph in the results section now contains the following statement:

Percolation is a transition in connectivity; rigidity percolation is a transition from elastic to floppy, with a dramatic peak in low-energy bosonic excitations we believe is common to strange metals.

Let us now further elaborate on the differences between standard percolation and rigidity transitions. In the standard “random-resistor network” picture of a percolative transition between a metal and an insulator, the network consists of links that represent either an insulator or a metal. Beyond a critical probability threshold for occupying the links, there is a transition from a metal to an insulator. While there is a superficial similarity between such a percolative metal-insulator transition and RP, there are two key differences that are important for our analysis.

- 1) Conceptually, the physical setting that RP captures in our analysis is distinct from that of geometric bond percolation, and is in a different universality class. The upper critical dimension for both jamming and RP is also believed to be 2 (while regular percolation has upper critical dimension of 6), which lends credence to a mean-field like approach.
- 2) RP gives us direct access to the large density of classical low-energy bosonic (vibrational) excitations at the rigidity transition, which become strongly damped

even at the lowest frequencies near the critical point. To clarify again the rationale for the analogy between RP and density response of strange metals, we begin by noting that the well-defined bosonic quasiparticles within RP are the phonons, which are unrelated to the phonons in strange metals, but are the analogue of the plasmon (or zero-sound in the absence of Coulomb interactions). These are found deep within the solid side of our phase diagram. Near the transition to loss of rigidity, which has parallels with “optimally doped” strange metals, these bosonic excitations are short-lived and contribute to what becomes an incoherent plateau in the density-density response. This is consistent with the experiment which detects this plateau, and no long-lived plasmons except at the smallest wavevector q . Finally, we stress that the microscopic mappings between the intermediate-energy bosonic excitations near RP and the density fluctuations in a strange metal remain unknown - we only seek to describe their phenomenology by comparing the experiment with that of a jamming-type transition.

(2) Related to point (1), in systems with jamming one can typically tune the transition by varying the particle density. What is the equivalent to a tunable parameter in the systems considered in the present work? If such tunability were achievable experimentally, I would expect that below the transition the system would show properties of an insulator or some other type of metal. Is there a reason that only a certain class of systems seems to fall just at the jamming transition, and can these systems be tuned?

We thank the referee for asking a great question. In the actual theoretical model being studied, the quantity being tuned is a microscopic “coordination number”. The elastic membrane through which the phonons propagate loses rigidity below a certain critical coordination z_c (or occupation probability p_c). We study the fate of the phonons, as manifested in the density response of this elastic membrane, across this transition. Remarkably, we have pointed out that this phenomenology is similar to the density response of a strongly correlated electronic system, where the quantity being tuned is a “dopant concentration”. Note that the latter changes the density of holes in the system, while simultaneously introducing sources of disorder. As we pointed out in response to Q1 above and in the manuscript, the exact quantitative mapping between z (p) and the dopant concentration is unclear and remains an exciting direction to pursue in future work.

Based on the surprising conceptual analogies between these two pictures and the detailed comparison between theory and experiment, we suggest that a bosonized degree of freedom associated with the underlying electronic degrees of freedom undergoes a rigidity-type transition at intermediate energies. However, this does NOT imply that the underlying

electronic degrees of freedom are undergoing a metal-insulator transition across the rigidity percolation.

We have now added a sentence to the caption for Fig. 1a:

We hypothesize that the two-particle density response over a broad range of intermediate energies near the hole-doping induced “transition” associated with the electrons near optimal doping in cuprates [15] can be described as a rigidity-type transition.

(3) Jamming and the glass transition actually occur at very different temperatures and densities in particle like systems. Is this also the case in this system?

The jamming transition is a zero-temperature critical point. The glass transition can be viewed as a transition (or, in finite dimensions, perhaps a crossover) emerging from the zero-temperature classical critical point. The large density of harmonic excitations at intermediate energies in glasses (the boson peak) is attributed to the large density seen in jamming, while glasses at lower temperatures and energies differ substantially from jamming. Jamming in glasses is a kind of avoided critical point, where attractive forces are a relevant parameter causing the long-wavelength, low frequency behavior to change but leaving the intermediate frequency behavior of jamming criticality. We are making a similar analogy. Our rigidity transition in strange metals is a zero-temperature classical critical point, where quantum fermionic fluctuations are relevant. At low energies, electrons and Fermi surfaces and quantum physics dominate the behavior, but at short wavelengths and intermediate frequencies the bosonized electron degrees of freedom retain the behavior at the rigidity transition. By leveraging our quantitative theory of the vibration modes at the rigidity transition, we can make this analogy precise and test it.

(4) One thing that I think could make this picture a bit different is that charge states are also coupled to a substrate that can provide quenched disorder. It is known that quenched disorder or pinning can change the jamming transitions by adding a characteristic cutoff length or changing the location of the transition points. It might be possible that if there is additional quenched disorder, some parts of the system will naturally be near the jamming transition, or there could be an extension of the regime where criticality associated with jamming can occur. The authors should add some comments on this and the possible role of disorder or heterogeneity in these samples and how jamming can still work. Additionally, since such disorder combined with jamming or clogging can produce strong memory effects, do these strange metals also have memory effects or fragility? This might be a good prediction as well. Some related papers on the particle systems are:

Jamming in systems with quenched disorder

C. J. Olson Reichhardt, E. Groopman, Z. Nussinov, and C. Reichhardt

Phys. Rev. E 86, 061301 (2012)

Pinning Susceptibility: The Effect of Dilute, Quenched Disorder on Jamming

Amy L. Graves, Samer Nashed, Elliot Padgett, Carl P. Goodrich, Andrea J. Liu, and James P. Sethna

Phys. Rev. Lett. 116 235501 (2016)

Crossover from Jamming to Clogging Behaviours in Heterogeneous Environments

H. Péter, A. Libál, C. Reichhardt, and C. J. O. Reichhardt

Sci Rep 8 10252 (2018)

Jammed solids with pins: Thresholds, force networks, and elasticity

Andy L. Zhang, Sean A. Ridout, Celia Parts, Aarushi Sachdeva, Cacey S. Bester, Katharina Vollmayr-Lee, Brian C. Utter, Ted Brzinski, and Amy L. Graves

Phys. Rev. E 106 034902 (2022)

We thank the referee for this suggestion. In fact, in our previous version of the manuscript, we had already pointed out that in order to get a better qualitative agreement between the strange metal experiments and our theoretical analysis for RP, we smear the density response over a long-wavelength disorder distribution. Thus, even when one is an average distance away from the critical point, the smearing leads to the effect of including contributions from “mesoscopic scale” rigid, floppy and critical regions, respectively.

To clarify this further, we have now added on page 4 a paragraph discussing our averaging over inhomogeneity, noting the possible important effects of pinning that we are neglecting, and adding the suggested citations:

We thus assume that the variations in doping level change the ‘distance’ to the onset of rigidity. Our averaging presumes the disorder does not couple to the translational Goldstone mode of the transition (by solely changing the density of bonds). The doping, however, breaks translational symmetry, and pinning on defects is also known to lower the threshold of rigidity [41-44]. Adding the effects of pinning to our analysis could be fruitful in future work.

(5) Fig. 2 is a nice diagram explaining the different decay exponents, but I think another nice diagram to include would be something like a jamming phase diagram but for the metal case. In that case, what is being tuned, the doping or charge density, the temperature,

or other quantities? This would probably get some more interest in the paper from the soft matter community.

We thank the referee for this interesting suggestion. As already noted in our response to the referee's previous Q1 and Q2, a microscopically accurate picture of how "jamming" arises in a metal (i.e. starting with an electronic model with a conserved density) remains presently unclear. Therefore, we prefer not to speculate on and draw a potentially misleading diagram directly for the metal. This clearly remains an interesting problem for the future that is beyond the scope of the present study.

However, we reiterate here our conjectured connection between RP and strange metals, which is now included in the updated caption of Fig. 1:

We hypothesize that the two-particle density response over a broad range of intermediate energies near the hole-doping induced "transition" associated with the electrons near optimal doping in cuprates can be described as a rigidity-type transition.

Overall the connection to soft matter is nice but I would like to know why these materials seem to go to the critical point. The authors could add some comments on this point. For example, it could be due to longer range correlations of the charge that allow the system to be close to some kind of jamming, as found for dislocations:

Dislocations Jam at Any Density

Georgios Tsekenis, Nigel Goldenfeld, and Karin A. Dahmen

Phys. Rev. Lett. 106, 105501 (2011)

This is an interesting question. It is plausible that certain strange metals with a short electronic mean-free path are so strongly correlated (driven in part by their proximity to a parent Mott insulator), that the electronic fluid becomes nearly jammed. Furthermore, the doping-induced inhomogeneity might further increase the propensity towards jamming. We have added a reference to Tsekenis et al. to our 'umbrella of jamming' discussion on page 2. However, it is far from being obvious (and is possibly quite unlikely) that our model 'self-organizes' at the critical point as was pointed out in this interesting paper.

(6) Since the authors use arguments from jamming, could this work imply the presence of some other phenomena that is found in jamming, such as the situation where an already jammed system could unjam under a drive, undergo thermal unjamming, or so forth? As I mentioned already, jammed systems can in many cases have memory or fragility effects, particularly near the threshold.

We thank the referee for an excellent suggestion. Indeed, if the electronic fluid over a range of energy scales looks nearly “jammed”, there will likely be memory effects and interesting relaxation timescales associated with this state. However, it is presently unclear how an external probe that couples to the electronic density can help reveal these interesting features. We have now added a penultimate sentence speculating on these lines, saying;

A broad implication of our hypothesis is that over a range of intermediate energy scales over which the density correlations in strange metals appear to display features like jamming, the electronic fluid might also display interesting memory effects known to arise in glassy systems and near rigidity transitions. Finding new experimental ways to probe this physics remains an interesting future direction.

Reviewer #2 (Remarks to the Author):

I very strongly recommend the publication of the manuscript "Jamming and Unusual Charge Density Fluctuations of Strange Metals" by Thornton, Liarte, Abbamonte, Sethna, and Chowdhury. This article is very well written and motivated. The presented analysis is extremely straightforward and lucid. There are no real required changes at this end prior to publication.

We thank the reviewer for their thoughtful reading of our manuscript and for recommending publication.

As an extremely minor point, that should not be misconstrued as a condition for acceptance, it might be remarked that under the noted "umbrella of jamming", Ref. [44] already earlier suggested, from a rather different vantage point, the prospect of jamming in interacting electronic systems. Indeed, similarly inspired by the phenomenology of the high temperature superconductors, Ref. [44] illustrated how "self-organized" jamming may be triggered by particle interactions.

We thank the referee for bringing this up. Indeed, based on our understanding, Ref. [44] (<https://journals.aps.org/prb/abstract/10.1103/PhysRevB.87.184202>) pointed out intriguing connections between the non-equilibrium dynamics of classical Fokker-Planck systems and zero-temperature quantum systems. Specifically, the paper takes critical exponents associated with linear soft sphere jamming and posits a quantum system with bosons of very large mass that should undergo a jamming-like transition at zero temperature (note their Fig. 1 is for a quantum system at $T=0$). They note that a similar dynamical exponent to the one they predict through their classical-quantum correspondence was found in a

model for a Bose glass.

We have now added the following phrase in the sentence where we discuss other examples of jamming (i.e. under the “umbrella of jamming”)

... and analogues of metal-insulator transitions for interacting quantum Bosons [31].

We have also added the following sentence at the end of the paper

In this regard, exploring possible connections between the low-energy vibrational excitations near jamming and the low-energy non-quasiparticle-like excitations in the solvable quantum Sachdev-Ye-Kitaev models [10,47] will be an interesting theoretical exercise.

The manuscript by Thonton, Liarte, Abbamonte, Sethna, and Chowdhury advances a conceptually neat idea and simple framework that may rationalize the phenomenology of bad metals. This is important and may be appreciated by numerous readers.

We appreciate the positive remarks by the referee.

REVIEWERS' COMMENTS

Reviewer #1 (Remarks to the Author):

In any case I can now recommend for publication and think the authors did a nice job in addressing all my points. The other referee has already agreed for publication. As I said in my initial report that although there has been some vague attempts at linking jamming to hard condensed matter systems this is the first work in my mind that makes very clear case, predictions and picture for how it actually could be achieved. As such I think this paper could very impactful.